# Association between COVID-19 and Postoperative Neurological Complications and Antipsychotic Medication Use after Cancer Surgery: A Retrospective Study

**DOI:** 10.3390/jpm13020274

**Published:** 2023-01-31

**Authors:** Juan P. Cata, Jian Hu, Lei Feng, Caroline Chung, Scott E. Woodman, Larissa A. Meyer

**Affiliations:** 1Department of Anesthesiology and Perioperative Medicine, The University of Texas—MD Anderson Cancer Centre, Houston, TX 77030, USA; 2Anesthesiology and Surgical Oncology Research Group, Houston, TX 77030, USA; 3Department of Cancer Biology, The University of Texas—MD Anderson Cancer Centre, Houston, TX 77030, USA; 4Department of Biostatistics, The University of Texas—MD Anderson Cancer Centre, Houston, TX 77030, USA; 5Department of Radiation Oncology, The University of Texas—MD Anderson Cancer Centre, Houston, TX 77030, USA; 6Data-Driven Determinants for COVID-19 Oncology Discovery Effort (D3CODE) Team, University of Texas—MD Anderson Cancer Centre, Houston, TX 77030, USA; 7Department of Genomic Medicine, The University of Texas—MD Anderson Cancer Centre, Houston, TX 77030, USA; 8Department of Gynecology Oncology and Reproductive Medicine, The University of Texas—MD Anderson Cancer Centre, Houston, TX 77030, USA

**Keywords:** cancer, COVID-19, complications, surgery

## Abstract

Introduction: Millions of Americans infected with the severe acute respiratory syndrome-associated coronavirus-19 (COVID-19) need oncologic surgery. Patients with acute or resolved COVID-19 illness complain of neuropsychiatric symptoms. How surgery affects postoperative neuropsychiatric outcomes such as delirium is unknown. We hypothesize that patients with a history of COVID-19 could have an exaggerated risk of developing postoperative delirium after undergoing major elective oncologic surgery. Methods: We conducted a retrospective study to determine the association between COVID-19 status and antipsychotic drugs during postsurgical hospitalization as a surrogate of delirium. Secondary outcomes included 30 days of postoperative complications, length of stay, and mortality. Patients were grouped into pre-pandemic non-COVID-19 and COVID-19-positive groups. A 1:2 propensity score matching was used to minimize bias. A multivariable logistic regression model estimated the effects of important covariates on the use of postoperative psychotic medication. Results: A total of 6003 patients were included in the study. Pre- and post-propensity score matching demonstrated that a history of preoperative COVID-19 did not increase the risk of antipsychotic medications postoperatively. However, respiratory and overall 30-day complications were higher in COVID-19 individuals than in pre-pandemic non-COVID-19 patients. The multivariate analysis showed that the odds of using postoperative antipsychotic medication use for the patients who had COVID-19 compared to those who did not have the infection were not significantly different. Conclusion: A preoperative diagnosis of COVID-19 did not increase the risk of postoperative antipsychotic medication use or neurological complications. More studies are needed to reproduce our results due to the increased concern of neurological events post-COVID-19 infection.

## 1. Introduction

As of 18 January 2023, the Centre for Disease Control has reported more than 101 million cases and over one million deaths of coronavirus disease 19 (COVID-19) in the United States (https://covid.cdc.gov/covid-data-tracker/#datatracker-home, accessed on 4 September 2022). Coronaviruses are common causes of respiratory tract illnesses, and COVID-19 is no exception [1]. Notably, the virus has the capability to damage many organ systems, including the central nervous system. A growing level of evidence indicates that patients infected with COVID-19 report neurological symptoms that can extend for several months (long COVID) after the acute phase of the infection [2]. Furthermore, patients with mild COVID-19 illness may develop acute or chronic neuropsychiatric symptoms, including confusion, delirium, and memory problems [2,3,4]. In fact, neuropsychiatric manifestations of COVID-19 have been reported in up to 85% of the patients in the acute or subacute phase of the disease [2]. A meta-analysis revealed a similar rate (80%) of long-lasting neuro-psychiatric symptoms, including fatigue (58%), headache (44%), and attention disorders (27%) [5]. Although not clearly determined, studies suggest that neuropsychiatric symptoms associated with COVID-19 are related to neuroinflammation, endothelial dysfunction, and neuronal injury [6]. Therefore, it has been recommended that COVID-19 could worsen the prognosis of patients in which neuroinflammation and immune dysregulation are present (i.e., major oncologic surgery) [6].

Clinical neurological and psychiatric syndromes are common in patients undergoing major cancer surgery. Delirium is a frequent (8–46%) complication and represents the extreme spectrum of the so-called postoperative cognitive disorders [7,8]. Patients with postoperative delirium have an increased risk of developing other postoperative complications, hospital readmission, and a higher healthcare-associated cost and mortality rate than those without delirium [9,10,11]. Precipitating factors of postoperative delirium include age, duration of surgery, the extent of surgery, and perioperative medication administration, such as high doses of opioids and benzodiazepines [12,13,14]. However, patients with a diminished brain reserve, such as those with dementia, also have an increased risk for postoperative delirium [7,14,15]. Studies have shown Alzheimer’s disease-like features in patients with a history of long COVID, suggesting permanent brain damage [16].

Patients with a history of COVID-19 infection may have a higher risk of complications and mortality postoperatively [17]. However, recent studies have suggested the opposite [18,19,20]. Additionally, pooled data from 24 studies that included 6762 patients did not show a significant increase in complications in oncological patients requiring surgery during the pandemic [21]. However, no studies have investigated the association between a history of COVID-19 and postoperative delirium or neurological complications. We hypothesized that patients with a history of COVID-19 disease or infection would have an exaggerated risk of developing postoperative delirium or cognitive dysfunction than pre-pandemic non-COVID-19 patients [22]. To test this hypothesis, we conducted a retrospective study to predict whether a preoperative history of COVID-19 disease or infection increased the risk of postoperative antipsychotic medication use and neurological complications.

## 2. Methods and Materials

This study was approved by the Institutional Review Board (IRB) “The University of Texas—MD Anderson Cancer Center” (1 January 2021). The IRB granted a waiver of informed consent to the investigators. This study was conducted according to MD Anderson Cancer Center institutional guidelines and following the Declaration of Helsinki ethical principles for medical research involving human subjects. Patients who had cancer surgery between 1 January 2019 and 31 August 2021 were included in this study if they were >18 years of age and stayed in the hospital for at least 48 h after surgery. We excluded ambulatory and emergency surgeries, and patients with repeated surgeries. A positive history of COVID-19 was defined as a positive polymerase chain reaction (PCR) test any time before oncologic surgery or documented post-COVID-19 status as a disease in the electronic medical records [23]. Following the methodology used by Deng et al., we considered no COVID-19 patients as those who had surgery before 31 December 2019, as these presented no history of COVID-19 [24]. It is worth mentioning that from the pandemic’s beginning until today, patients will only undergo scheduled surgery if two weeks have passed after PCR positivity, there is confirmation of a negative PCR or in the absence of symptoms. Therefore, none of the patients in the analysis had “acute” symptoms of COVID-19.

Clinical data, including demographics, American Society Anesthesiologists’ (ASA) Physical Status, history of cancer treatments, primary surgical service, postoperative complications, and mortality, obtained from the electronic medical record system were used to analyze integration into a cloud-based tool within the Context Engine at the University of Texas—MD Anderson’s Cancer Centre data [25,26]. The primary outcome was the rate of postoperative administration of antipsychotic drugs (haloperidol, risperidone, quetiapine, olanzapine, ziprasidone, paliperidone, aripiprazole, and clozapine) during postoperative hospitalization since these medications are typically given to patients with positive features (i.e., agitation) of delirium [27]. Secondary outcomes included length of stay, 30-day neurological (cerebrovascular accident or stroke), cardiovascular (congestive heart failure, deep vein thrombosis, hypertension, myocardial infarction), and respiratory (pneumonia, pulmonary hypertension, and pulmonary embolism) postoperative complications, and overall 30-day morbidity and 30-day mortality.

### Statistical Analysis

Patient demographics, preoperative cancer treatment, primary surgical service, anesthesia duration, and postoperative clinical outcomes were summarized through descriptive statistics. The chi squared test or Fisher’s exact test was used to evaluate the association between the patient demographical/clinical characteristics and the use of postoperative antipsychotic medication or postoperative complications. The Wilcoxon rank-sum test was used to assess the difference in a continuous variable between the COVID-19 history and no COVID-19 history groups, between the use of postoperative antipsychotic medication and no use of postoperative antipsychotic medication groups, or between the postoperative complications and no complications groups. The baseline covariates with *p*-values < 0.3 from the above tests were included as covariates when fitting multivariable logistic regression models to assess their effects on the use of postoperative antipsychotic medications or complications. We then conducted a propensity score matching analysis to adjust for selection bias in this observational study. The standardized difference was calculated to evaluate the balance in baseline covariates after PSM between the two groups of patients. We used 0.4 as the significance level for a baseline covariate to stay in the multivariable logistic regression model when using the post-propensity score matching dataset to assess covariate effects on the use of postoperative antipsychotic medication or complications.

Approximately 28,700 patients were part of the D3CODE registry at the time of the study. Among them, about 600 patients had a history of COVID-19 before surgery, given the prevalence rate of COVID-19 of 2.1%. The primary endpoint was the status of postoperative delirium during hospitalization recorded in patient medical notes and/or indicated by the use of antipsychotic medication. We estimated a rate of postoperative delirium for patients without a COVID-19 history of 30%. Then, the study would have 81% power to detect a difference in the rate of postoperative delirium between the patients with and without COVID-19 history when 35.5% of the patients with COVID-19 history would suffer postoperative delirium, assuming a two-sided type I error rate of 0.05 (nQuery Advisor 7.0). A *p* < 0.05 was considered statistically significant. Statistical software SAS 9.4 (SAS, Cary, NC, USA) was used for all the analyses.

## 3. Results

A total of 6003 patients who met the inclusion criteria were included in the study. Of them, 373 subjects tested positive or had a history of COVID-19 disease from 1 January 2020 until 31 August 2021. Most patients were ASA physical status 3, received chemotherapy, and did not have radiation. Most subjects were grouped in the surgical oncology service (i.e., abdominal surgeries), while fewer had gynecological oncology surgeries (Table 1).

Overall, 167 patients (2.8%) used postoperative antipsychotic medications, 143 patients (2.4%) experienced neurological complications, 739 patients (12.3%) experienced cardiovascular complications, 1060 patients (17.7%) experienced respiratory complications, and 1628 patients (27.1%) experienced at least one of the above postoperative complications (Table 2).

Before propensity score matching, patients with a history of COVID-19 were slightly but statistically significantly younger than pre-pandemic non-COVID-19 subjects (Table 1). Other demographic and treatment-related variables were not significantly different (Table 1). The univariate analysis demonstrated that postoperative psychotic medications (*n* = 160, 2.8% vs. *n* = 7, 1.9%) and neurological (*n* = 136, 2.4% vs. *n* = 7, 1.9%) complications were not statistically different between both groups of interest. However, there was a significantly higher rate of postoperative pulmonary complications in patients with a history of COVID-19 (*n* = 81, 21.7%) than in pre-pandemic non-COVID-19 subjects (*n* = 979, 17.4%, *p* = 0.033). Length of stay and 30 days mortality rate were nearly identical in both groups of patients (Table 2).

With the adjustment of age, gender, race, ASA physical status, history of radiation, and duration of anesthesia in a multivariable logistic regression model, the estimated odds of using postoperative antipsychotic medication in patients who had a history of COVID-19 compared to the patients who pre-pandemic non-COVID-19 individuals was not statistically significantly different (OR = 0.65, 95% CI = 0.30, 1.40; *p* = 0.27) (Table 3). With the adjustment of age, gender, race, ASA physical status, history of radiation, and duration of anesthesia, the odds of having postoperative respiratory complications for the patients who had a history of COVID-19 were 34% higher compared to the patients who did not have a history of COVID-19 (OR = 1.34, 95% CI = 1.04, 1.73; *p* = 0.026) (Table 4).

### Matched Patient Population

To reduce the bias due to confounding variables, we conducted propensity score matching. For this, 1119 patients were included in the analysis. A total of 373 patients with a preoperative diagnosis of COVID-19 were 1:2 matched to 746 pre-pandemic non-COVID-19 patients by age at surgery, gender, and primary service. The matched population’s mean age (standard deviation) was 55.21 (13.94) years. Moreover, there were slightly more women (*n* = 592, 52.9%) than males (*n* = 527, 47.1%). The standardized differences for all covariates were <6% in the post-matching dataset, suggesting a substantial bias reduction between the two groups. As shown in Table 1, demographic, preoperative cancer treatments, primary surgical service, and anesthesia duration were not statistically significantly different between the patients with and without a history of COVID-19.

The use of antipsychotic medications as a surrogate for postoperative delirium was not statistically significantly different between COVID-19 (*n* = 7, 1.9%) and pre-pandemic non-COVID-19 (*n* = 20, 2.7%) patients (Table 2). In addition, postoperative neurological events, in which delirium was included, and cardiovascular complications were not statistically significantly different between both groups of patients. However, respiratory complications were statistically significantly higher in COVID-19 (*n* = 81, 21.7%) than in pre-pandemic non-COVID-19 (*n* = 121, 16.2%, *p* = 0.024) patients. Similarly, the presence of any complication 30 days after surgery was statistically significantly higher in patients with a history of COVID-19 (*n* = 112, 30%) than in pre-pandemic patients (*n* = 180, 24.1%, *p* = 0.034). Lastly, the length of stay in the hospital and 30-day mortality were not statistically significantly different between the two groups of patients.

Multivariable logistic regression models were fitted to estimate the effects of important covariates on the use of postoperative antipsychotic medication or postoperative respiratory complications. With the adjustment of gender and race, the effect of having a history of COVID-19 on the use of postoperative antipsychotic medication was a statistically independent predictor (OR = 0.68, 95% CI = 0.29, 1.63; *p* = 0.39) (Table 3). With the adjustment of age and gender, the odds of having postoperative respiratory complications for the patients who had a history of COVID-19 were 42% higher compared to the patients who did not have a history of COVID-19 (OR = 1.42, 95% CI = 1.04, 1.73; *p* = 0.028) (Table 4).

## 4. Discussion

With an increase in the population of COVID-19-infected patients requiring major surgery to treat solid malignancies, postoperative complications are likely to increase mainly in unvaccinated individuals [28,29]. Previous studies have investigated the impact of acute or resolved COVID-19 infection on complications after cancer surgery, with pulmonary complications being the most commonly reported [30,31]. However, there is little evidence of the effects of preoperative COVID-19 illness on other postsurgical events, such as postoperative delirium or neuropsychiatric complications. Patients with COVID-19 can present neuropsychiatric symptoms (i.e., fatigue, headaches, or cognitive difficulties) even weeks after the acute phase of the infection [32] when surgery may be needed. In older hospitalized patients, delirium is the most common neurologic manifestation of COVID-19 and is associated with high mortality [33]. Our study focused on the use of antipsychotic drugs as a surrogate of postoperative delirium or neurological complications in patients with resolved infection. Our work did not show any significant association suggesting an increase in the rate of antipsychotic drug use or neurological complications in patients with a history of COVID-19 infections compared to similar patients without a history of COVID-19. However, our period of observation was limited to 30 days after surgery. Hence, we missed delirium or other cognitive disorders occurring after a month of surgery.

There are at least two explanations for our findings. First, the rates of antipsychotic drug use and neurological complications were low in pre-pandemic patients and those with a history of COVID-19 infection. Thus, we might have been statistically underpowered to demonstrate a clinically relevant or statistically significant difference in both outcomes, or an actual clinical difference does not exist. Second, we analyzed data from patients with a history of COVID-19 infection who were PCR-negative at the time of surgery and had most likely recovered from any COVID-19-like symptoms. In other terms, surgeries included in our analysis took place on highly selected patients, typically 4–8 weeks after the infection, who might have fully recovered from the infection, and with minimal or no COVID-19 sequela in the neurological system [34]. Thus, patients with any neuropsychiatric disorder or still symptomatic from the infection might not have undergone surgery during the period of our study or might have received different cancer treatments.

Our study suggests that surgery performed near the time of COVID-19 infection is associated with an increased risk of developing postoperative respiratory complications [24]. We observed a rate of 21.7% of respiratory complications in patients with a history of COVID-19. Using an administrative database, two studies reported a lower rate (1.4–4.8%) of respiratory complications than in our study [24,35]. However, differences in both studies can explain the dissimilar findings. For instance, in Deng et al.’s work, the investigators included patients with and without cancer, making it difficult to compare our results [24]. A meta-analysis indicated that postoperative pulmonary complications are the leading cause of death in patients with COVID-19 [36]. Doglietto et al. also reported a higher 30-day postoperative mortality rate for COVID-19 patients than non-COVID-19 individuals [36]. In fact, it has been estimated a 54.8% of excess postoperative deaths during the pandemic were attributable to COVID-19 [37]. While we observed a higher rate of postoperative pulmonary and overall complications in patients recovering from COVID-19 than those undergoing similar surgeries before the pandemic, the mortality rate did not show statistically significant differences. We can speculate that in our cohort of patients, postoperative pulmonary complications might have occurred in individuals who fully recovered from respiratory symptoms associated with COVID-19 and were asymptomatic at the time of surgery. Thus, the severity of the complication was milder than in patients with active or symptomatic infections, as demonstrated in a previous study [38]. This could also explain why the length of hospitalization was not statistically different between the two groups of patients.

In our work, cardiovascular complications occurred in 12.1% of COVID-19 patients. While the rate was slightly higher than in non-COVID-19 subjects (10.3%), it did not reach statistical significance. Bryant et al. reported the same rate of events in a recently published study that included 3997 patients [19]. Notably, the authors reported that risk decreased from the time of COVID-19 diagnosis to surgery [19]. We also investigated 30-day mortality rates in patients with and without a history of the infection. As reported by Quinn et al. in a large cohort of Italian patients, a prior history of COVID-19 infection did not increase the risk of postoperative mortality [20].

Our study has several limitations. First, this retrospective study used data from a cloud-based tool with inherent biases [25,26]. In addition, we included a cohort of patients who were mainly recovering or fully recovered from COVID-19 and undergoing elective oncologic resections at a high-volume comprehensive cancer center which may not be representative or generalizable across all settings. Although a recent study reported that the rate of postoperative complications is not increased in patients undergoing mastectomy in the ambulatory setting, our results cannot be generalized to oncological patients undergoing same-day surgery [39]. Second, we excluded patients undergoing emergency surgery or those with an active infection whose risk of postoperative delirium could have been higher than those recovering from the disease [40]. Unfortunately, our medical records at the time of the early stages of the pandemic did not differentiate between PCR (+) and a history of the disease. Therefore, we could not discriminate how many patients had the disease, and how many had only had a positive test. Third, our findings are also limited to the lack of preoperative clinical or imaging studies to determine the presence of cognitive dysfunction before surgery and a complete description of the postoperative neurological conditions after surgery, which can explain the lower-than-expected rates of neurological complications and use of antipsychotic medications [8]. Fourth, postoperative delirium was defined as administering new antipsychotic drugs during hospital admission. Due to the low rate of postoperative use of antipsychotic medications and neurological complications, it is possible that delirious patients were not adequately identified in our database or that it happened beyond 30 days after surgery. In addition, we might have missed patients with hypoactive delirium to whom antipsychotic medications were not prescribed and whose symptoms were not recorded in medical records. Lastly, our study included patients infected with early variants of the COVID-19 virus. Since September 2021, new virus variants have emerged; therefore, our results are not generalizable to patients with the Delta or Omicron variants. While we did not include a history of vaccination in our analysis, a large cohort study demonstrated no increase in major postoperative cardiovascular events or death in patients with or without COVID-19 vaccination [20].

In conclusion, oncologic patients with a preoperative diagnosis of COVID-19 did not show an association with neurological complications or antipsychotic medications as a surrogate of postoperative delirium. However, it was significantly associated with an increased risk of postoperative respiratory complications. More studies are needed to reproduce our results due to the increased concern of cognitive outcomes post-COVID-19 infection.

## Figures and Tables

**Table 1 jpm-13-00274-t001:** Demographic variables and endpoints.

Variable	Levels	Pre-Matching			Post-Matching		
		Pre-Pandemic (*n* = 5630)	COVID-19-Positive (*n* = 373)	*p*-Value	Pre-Pandemic (*n* = 746)	COVID-19-Positive (*n* = 373)	*p*-Value
Age, years	Mean (SD)	58.63 (14.05)	55.61 (14.03)	<0.0001	55.21 (13.94)	55.61 (14.03)	0.614
ASA physical status	1–2, *n* (%)	566 (10.1%)	37 (9.9%)	0.933	73 (9.8%)	37 (9.9%)	0.943
	3–5, *n* (%)	5064 (89.9%)	336 (90.1%)		673 (90.2%)	336 (90.1%)	
Gender	F, *n* (%)	2837 (50.4%)	195 (52.3%)	0.48	397 (53.2%)	195 (52.3%)	0.766
	M, *n* (%)	2793 (49.6%)	178 (47.7%)		349 (46.8%)	178 (47.7%)	
Race	Non-white, *n* (%)	1226 (21.9%)	81 (21.8%)	0.94	185 (25%)	81 (21.8%)	0.234
	White, *n* (%)	4362 (78.1%)	291 (78.2%)		555 (75%)	291 (78.2%)	
History of radiotherapy	No, *n* (%)	3616 (64.2%)	224 (60.1%)	0.103	468 (62.7%)	224 (60.1%)	0.384
	Yes, *n* (%)	2014 (35.8%)	149 (39.9%)		278 (37.3%)	149 (39.9%)	
Anticancer systemic therapies	Chemotherapy, *n* (%)	2593 (85.6%)	199 (88.1%)	0.429	344 (84.9%)	199 (88.1%)	0.544
	Chemo + immuno, *n* (%)	66 (2.2%)	2 (0.9%)		7 (1.7%)	2 (0.9%)	
	Immunotherapy, *n* (%)	369 (12.2%)	25 (11.1%)		54 (12.2%)	25 (11.1%)	
Surgery service	Gynecology, *n* (%)				454 (8.1%)	29 (7.8%)	0.236
	Head & neck, *n* (%)				783 (13.9%)	43 (11.5%)	
	Neurosurgery, *n* (%)				654 (11.6%)	43 (11.5%)	
	Orthopedic, *n* (%)				255 (4.5%)	28 (7.5%)	
	Plastic surgery, *n* (%)				403 (7.2%)	27 (7.2%)	
	Surgical oncology, *n* (%)				1312 (23.3%)	94 (25.2%)	
	Thoracic, *n* (%)				823 (14.6%)	49 (13.1%)	
	Urology, *n* (%)				946 (16.8%)	60 (16.1%)	
Anest. duration (min)	Mean (SD)	402.3 (188)	402.6 (188)	0.91	415.8 (190.1)	402 (189.1%)	0.2

ASA: American Society of Anesthesiologists. Chemo: Chemotherapy. Immuno: Immunotherapy.

**Table 2 jpm-13-00274-t002:** Postoperative complications, use of antipsychotics, and mortality.

Variable	Levels	Pre-Matching			Post-Matching		
		Pre-Pandemic (*n* = 5630)	COVID-19- Positive (*n* = 373)	*p*-Value	Pre-Pandemic (*n* = 746)	COVID-19- Positive (*n* = 373)	*p*-Value
Use of antipsychotic	No, *n* (%)	5470 (97.2%)	366 (98.1%)	0.272	726 (97.3%)	366 (98.1%)	0.408
medications (inpatient)	Yes, *n* (%)	160 (2.8%)	7 (1.9%)		20 (2.7%)	7 (1.9%)	
Postoperative inpatient	No, *n* (%)	5494 (97.6%)	366 (98.1%)	0.508	732 (98.1%)	366 (98.1%)	1
neurological complications	Yes, *n* (%)	136 (2.4%)	7 (1.9%)		14 (1.9%)	7 (1.9%)	
Postoperative inpatient	No, *n* (%)	4936 (87.7%)	328 (87.9%)	0.881	667 (89.4%)	328 (87.9%)	0.458
cardiovascular complications	Yes, *n* (%)	694 (12.3%)	45 (12.1%)		79 (10.6%)	45 (12.1%)	
Postoperative inpatient	No, *n* (%)	4561 (82.6%)	292 (78.3%)	0.033	625 (83.8%)	292 (78.3%)	0.024
respiratory complications	Yes, *n* (%)	979 (17.4%)	81 (21.7%)		121 (16.2%)	81 (21.7%)	
Postoperative	No, *n* (%)	4114 (73.1%)	261 (70%)	0.192	566 (75.9%)	261 (70%)	0.034
complications 30 days	Yes, *n* (%)	1516 (26.9%)	112 (30%)		180 (24.1%)	112 (30%)	
Length of stay (days)	Mean (SD)	5.32 (5.74)	5.06 (5.25)	0.289	5.34 (5.42)	5.06 (5.25)	0.27
Mortality (30 days)	No, *n* (%)	5613 (99.7%)	372 (99.7%)	1.000	745 (99.9%)	372 (99.7%)	1.000
	Yes, *n* (%)	17 (0.3%)	1 (0.3%)		1 (0.1%)	1 (0.3%)	

**Table 3 jpm-13-00274-t003:** Multivariable logistic regression for use of postoperative antipsychotics medications based on pre- and post-PSM datasets.

Covariate	Levels	Pre-PSM		Post-PSM	
		OR (95% CI)	*p*-Value	OR (95% CI)	*p*-Value
Gender	F vs. M			1.59 (0.72, 3.51)	0.2533
Race	Non-white vs. White	1.59 (1.03, 2.44)	0.0356	1.95 (0.67, 5.71)	0.2240
ASA physical status	3–5 vs. 1–2	1.51 (0.81, 2.81)	0.1914		
History of radiotherapy	Yes vs. No	1.37 (1.00, 1.87)	0.0498		
History of COVID-19	Yes vs. No	0.65 (0.30, 1.40)	0.2712	0.68 (0.29, 1.63)	0.3928
Anesthesia duration	≥5 h vs. <5 h	1.45 (1.02, 2.06)	0.0368		

**Table 4 jpm-13-00274-t004:** Multivariable logistic regression for postoperative respiratory complications based on before and after PSM datasets.

Covariate	Levels	Pre-PSM		Post-PSM	
		OR (95% CI)	*p*-Value	OR (95% CI)	*p*-Value
Age	≥60 vs. <60	1.37 (1.20, 1,58)	<0.0001	1.41 (1.03, 1.92)	0.0316
Gender	F vs. M	1.11 (0.97, 1.27)	0.1272	0.83 (0.61, 1.14)	0.2477
Race	Non-white vs. White	1.18 (1.00, 1.40)	0.0555		
ASA physical status	3–5 vs. 1–2	1.57 (1.21, 2.04)	0.0007		
History of radiotherapy	Yes vs. No	1.36 (1.19, 1.56)	<0.0001		
History of COVID-19	Yes vs. No	1.34 (1.04, 1.73)	0.0264	1.42 (1.04, 1.95)	0.0285
Anesthesia duration	≥5 h vs. <5 h	1.16 (1.00, 1.34)	0.0468		

## Data Availability

The material generated from this research belongs to the University of Texas—MD Anderson Cancer Center. The datasets used and/or analyzed during the current study are available from the corresponding author on reasonable request.

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
