# Peer review of "Association between COVID-19 and Postoperative Neurological Complications and Antipsychotic Medication Use after Cancer Surgery: A Retrospective Study"

_jpm, 2023, doi:10.3390/jpm13020274_

Round 1

Reviewer 1 Report

This is a very interesting retrospective trial conducted to determine the association between COVID-19 status and antipsychotic drugs during postsurgical hospitalization as a surrogate of delirium. 

The 6,003 patients sample included in the study showed that the use of postoperative antipsychotic medication use for the patients who had COVID-19 compared to those who did not have the infection were not significantly different. Moreover a preoperative diagnosis of COVID-19 did not increase the risk of postoperative antipsychotic medication use or neurological complications.

I have also minor queries :

- English language and style are fine/minor spell check required

- Provide in the Introduction and background part improvement of relevant references: Example Mental health in Italy after two years of COVID-19 from the perspective of 1281 Italian physicians: looking back to plan forward  Alessandro Cuomo , Mario Amore , Maria Felice Arezzo  3 , Sergio De Filippis  4 , Alessandra De Rose  3 , Silvestro La Pia  5 , Alessandro Pirani  6 , Riccardo Torta  7 , Andrea Fagiolini  8 Affiliations  expand PMID: 35948983  PMCID: PMC9363263  DOI: 10.1186/s12991-022-00410-5 Free PMC article

Author Response

We appreciate the Editor comment. We have expanded the manuscript to 3,000 words.

Reviewer 2 Report

The authors conducted a retrospective study with approximate six thousand included patients to explore the association between preoperative history of COVID-19 disease and risk of postoperative antipsychotic medication use and neurological complications, which is a novel and interesting topic. The authors performed decent analysis with the available patient data, and had a thorough discussion regarding study design and limitations. 

It need to be more careful to define or mention "history of COVID-19 infection". In introduction there is sentence "Patients with a history of COVID-19 infection have a higher risk of complications and mortality postoperatively[12]". While the patients included in that study are with perioperative SARS-CoV-2 infection. 

The simple word "COVID-19" in manuscript title may be misleading as it may suggest "active COVID-19 disease", better to use such as "COVID-19 history".

Author Response

NA